Atmospheric Measurement Techniques

# Measuring OVOCs and VOCs by PTR-MS in an urban roadside microenvironment of Hong Kong: relative humidity and temperature dependence, and field inter-comparisons

Long Cui<sup>1</sup>, Zhou Zhang<sup>2</sup>, Yu Huang<sup>3,4</sup>, Shun Cheng Lee<sup>1</sup>, Donald Ray Blake<sup>5</sup>, Kin Fai Ho<sup>6</sup>, Bei Wang<sup>7</sup>, 5 Yuan Gao<sup>1</sup>, Xin Ming Wang<sup>2</sup>, and Peter Kwok Keung Louie<sup>8</sup>

<sup>1</sup>Department of Civil and Environmental Engineering, The Hong Kong Polytechnic University, Hung Hom, Hong Kong, China <sup>2</sup>State Key Laboratory of Organic Geochemistry, Guangzhou Institute of Geochemistry, Chinese Academy of Sciences, Guangzhou, China

<sup>3</sup>Key Lab of Aerosol Chemistry & Physics, Institute of Earth Environment, Chinese Academy of Sciences, Xi'an, China
 <sup>4</sup>State Key Lab of Loess and Quaternary Geology (SKLLQG), Institute of Earth Environment, Chinese Academy of Sciences, Xi'an, China

<sup>5</sup>Department of Chemistry, University of California, Irvine, CA, USA

<sup>6</sup>School of Public Health and Primary Care, The Chinese University of Hong Kong, Shatin, Hong Kong, China

<sup>7</sup>Faculty of Science and Technology, Technological and Higher Education Institute of Hong Kong, Hong Kong, China
 <sup>8</sup>Hong Kong Environmental Protection Department, Revenue Tower, 5 Gloucester Road, Wanchai, Hong Kong, China
 *Correspondence to*: S.C. Lee (ceslee@polyu.edu.hk)

# Abstract.

Volatile organic compound (VOC) control is an important issue of air quality management in Hong Kong because ozone

- formation is generally VOC limited. Several oxygenated volatile organic compound (OVOC) and VOC measurement techniques, namely, (1) off-line 2,4-dinitrophenylhydrazine (DNPH) cartridge sampling followed by High Performance Liquid Chromatography (HPLC) analysis, (2) on-line gas chromatography (GC) with flame ionization detection (FID), and (3) off-line canister sampling followed by GC with mass spectrometer detection (MSD), FID, and (electron capture detection) ECD, were applied during this study. For the first time, the proton transfer reaction mass spectrometry (PTR-MS) technique was
- also introduced to measured OVOCs and VOC in urban roadside area of Hong Kong. The integrated effect of ambient relative humidity (RH) and temperature (T) on formaldehyde measurements by PTR-MS was explored in this study. A Poly 2D regression was found to be the best nonlinear surface simulation (r = 0.97) of the experimental reaction rate coefficient ratio, ambient RH and T for formaldehyde measurement. This correction method was found to be better than correcting formaldehyde concentrations directly via the absolute humidity of inlet sample, based on a two-year field sampling campaign
- at Mong Kok (MK) in Hong Kong. For OVOC species, formaldehyde, acetaldehyde, acetone and MEK showed good agreements between PTR-MS and DNPH-HPLC with with slopes of 1.00, 1.10, 0.76 and 0.88, respectively, and correlation coefficients of 0.79, 0.75, 0.60 and 0.93, respectively. Overall, fair agreements were found between PTR-MS and on-line GC-FID for benzene (slope = 1.23, r = 0.95), toluene (slope = 1.01, r = 0.96) and C<sub>2</sub>-benzenes (slope = 1.02, r = 0.96) after correcting benzene and C<sub>2</sub>-benzenes levels which could be affected by fragments formed from ethylbenzene. For the inter-
- comparisons between PTR-MS and off-line canister measurements by GC-MSD/FID/ECD, benzene showed good agreement with a slope of 1.05 (r = 0.62), though PTR-MS had lower values for toluene and C<sub>2</sub>-benzenes with slopes of 0.78 (r = 0.96) and 0.67 (r = 0.92), respectively. All in all, the PTR-MS instrument is suitable for OVOC and VOC measurements in urban roadside areas.

## **1** Introduction

Volatile organic compounds (VOCs), which are important precursors of tropospheric ozone and secondary organic aerosols (SOAs) (Sillman, 2002), can be emitted from multiple anthropogenic sources (e.g., vehicular emissions, industrial emissions and solvent usage) and biogenic sources (Atkinson and Arey, 2003; Watson et al., 2001). VOCs can also have adverse impact

30

on human beings (von Schneidemesser et al., 2010). As one of the most densely populated cities in the world, Hong Kong has over 7.2 million people and more than 699,540 registered vehicles in an area of 1,104 km<sup>2</sup> as of December in 2014 (Hong Kong Transport Department, 2014). Special attention has been paid on the characteristics of roadside VOCs and their impacts on the local air quality of Hong Kong during the past years (Louie et al., 2013; Ling and Guo, 2014; Ho et al., 2004; Lee et al.,

- 2002; Guo et al., 2007). Previous studies showed that vehicular emissions is one of the major contributors to ambient VOCs in Hong Kong (Guo et al., 2007). Lau et al. (2010) found that 31-48% of ambient VOCs in Hong Kong were generated by vehicle- and marine vessel-related sources in 2002-2003, and the percentage increased to 40-54% in 2006-2007. In order to investigate urban roadside VOCs in Hong Kong, multiple sampling and analytical techniques were used, such as off-line 2,4-dinitrophenylhydrazine (DNPH) cartridge sampling followed by High Performance Liquid Chromatography
- (HPLC) analysis for oxygenated volatile organic compounds (OVOCs), on-line gas chromatography (GC) with flame ionization detection (FID), and off-line canister sampling followed by GC with mass spectrometer detection (MSD), FID, and (electron capture detection) ECD for VOCs (Ho et al., 2013; Cheng et al., 2014; Ou et al., 2015). These techniques, however, can be impacted by their relatively low sampling resolution that may lead to the underestimation or overestimation of ambient OVOC and VOC concentration levels (Jobson et al., 2010; Wang et al., 2014; Wisthaler et al., 2008). Moreover, since there is
- no single technique that can measure all OVOC and VOC species simultaneously, so different kinds of sampling and analytical techniques are usually used to obtain the measurements of these species (Jobson et al., 2010; Warneke et al., 2011a; Ambrose et al., 2010; Wisthaler et al., 2008).

Proton transfer reaction-mass spectrometry (PTR-MS) is a relatively novel method that can fulfill on-line measurements of OVOCs and VOCs at trace levels in ambient air. Proton transfer enables soft ionization of chemical species that have a higher

- proton affinity (PA) than that of the reagent species (i.e. H<sub>2</sub>O). PTR-MS does not need any sample treatment such as drying and/or precondensation like gas chromatographic analysis, and it is available for monitoring OVOCs which are quite difficult to quantify from canister samples. Hence, PTR-MS has recently been widely used in atmospheric chemistry research (de Gouw and Warneke, 2007; Jobson et al., 2010).
- Formaldehyde (HCHO) is one of the most abundant OVOCs in Hong Kong. The lifetime of HCHO in the boundary layer is estimated to be only several hours when photolysis and its reaction with OH radical occur with sunlight. Because HCHO is carcinogenic (Kerns et al., 1983), it is one of the OVOC species of most interest in urban areas. Only off-line DNPH cartridge sampling analyzed by HPLC has previously been used to determine the level of ambient formaldehyde in Hong Kong (Louie et al., 2013; Cheng et al., 2014; Ho et al., 2002; Guo et al., 2004). PTR-MS was first introduced to measure ambient concentrations of formaldehyde continuously in this study. HCHO can be protonated as following reaction (Ra):

| (Ra)  | $\rm HCHO + H_{3}O^{+} \rightarrow \rm HCHO \cdot \rm H^{+} + \rm H_{2}O$ |  |  |  |  |
|-------|---------------------------------------------------------------------------|--|--|--|--|
| (R-a) | HCHO $\cdot H^+ + H_2O \rightarrow HCHO + H_3O^+$                         |  |  |  |  |

However, because the proton affinity (PA) of formaldehyde (170.4 kcal mol<sup>-1</sup>) is just slight higher than that of water (165.2 kcal mol<sup>-1</sup>), the backward reaction (R-a) of protonated formaldehyde with  $H_2O$  can lead to an underestimation of HCHO by PTR-MS. Jobson and McCoskey (2010) found that the sensitivity to HCHO can be improved by removing water vapor from

- 35 the air sampling inlet. Correction for absolute humidity effects on HCHO measurement by PTR-MS was further discussed by Vlasenko et al. (2010). However, the previous studies did not fully discuss the effect of ambient relative humidity (RH) and/or temperature (T) separately and integratively on HCHO measurements by PTR-MS. In addition, few studies have reported inter-comparisons between PTR-MS and DNPH-HPLC during field studies, especially in urban roadside areas. In this study, PTR-MS was applied in Hong Kong to measure ambient OVOCs and VOCs coupled with other OVOC and VOC
- 40 measurement techniques. The effect of ambient conditions (RH and T) on OVOC and VOC measurements by PTR-MS are discussed in this paper. Detailed comparisons of PTR-MS, DNPH-HPLC, on-line GC-FID and off-line canister sampling followed by GC-MSD/FID/ECD for OVOC and VOC measurements at an urban roadside site in Hong Kong are conducted.

#### 2 Methodology

## 2.1 Field sampling site

An urban roadside station, Mong Kok (MK) Air Quality Monitoring Station (AQMS), was chosen as the field sampling site in this study. This sampling site, which is one of the three roadside monitoring stations established by the Hong Kong Environment Protection Department (HKEPD), is located in a mixed commercial and residential area with heavy daily traffic

in Hong Kong. Briefly, MK AQMS is the best representative of the roadside environment in Hong Kong, and more detailed information can be found elsewhere (Chan et al., 2002; Lee et al., 2002). The field study was conducted in May 2013, August 2013, November 2013, February 2014, May 2014 and August 2014, to cover the seasonal variations of traffic-related pollutants.

#### 2.2 PTR-MS

- A commercially available PTR-MS instrument (IONICON Analytik GmbH, Innsbruck, Austria) was used in this study. PTR-MS instrument has been described in detail elsewhere (Lindinger et al., 1998; de Gouw and Warneke, 2007; de Gouw et al., 2003). Briefly, PTR-MS mainly consists of a discharge ion source, a drift tube and a quadrupole mass spectrometer. In this study, H<sub>3</sub>O<sup>+</sup> was utilized as the ion source, the drift tube was operated at 2.2 mbar pressure and the electric field was maintained at 600 V difference. The E/N (E stands for the electric field strength and N stands for the air density inside drift tube) value in
- the drift tube was kept at 136 Townsend (Td). The electric field maintains a controlled ion velocity in the drift tube so that the clustering of water ions can be reduced. The PTR-MS inlet system and the drift tube were maintained at 60 °C to minimize wall losses.

The PTR-MS instrument was located in a shelter with an air conditioning system at the MK station. The inlet was located about 2 m above the ground. A 1/8" Teflon line was used as a sample line. The sample air was pumped at a flow rate of 75 mL

20 min<sup>-1</sup>, with an estimated residence time of 2 s in the flow tube. An in-line particulate filter was used to prevent particles from entering the instrument.

Ionimed mix-VOCs gas standard (IONICON Analytik GmbH, Innsbruck, Austria) was used for PTR-MS calibration in this study. Seventeen species were contained in the gas standard, namely formaldehyde, methanol, acetonitrile, acetaldehyde, ethanol, acrolein, acetone, isoprene, crotonaldehyde, 2-butanone, benzene, toluene, o-xylene, chlorobenzene,  $\alpha$ -pine, 1,2-

- dichlorobenzene and 1,2,4-trichlorobenzene with mixing ratios about 1ppm. Standard VOC gas mixtures (Supelco TO-14 Calibration Mix) were used for determining the transmission curve of the PTR-MS instrument. Zero air was generated by a Gas Calibration Unit (GCU) (IONICON Analytik GmbH, Innsbruck, Austria) with a VOC-scrubber installed inside the GCU. The relative humidity and temperature of inlet gas were set by adjusting the humidification chamber and a dew point mirror inside the GCU. Ionimed mix-VOCs gas standard was diluted with four different ratios (0.08, 0.06, 0.04 and 0.001) to calibrate
- the PTR-MS. The accuracy of the PTR-MS was 20%, and the measurement precision was about 10%. Calibrations were done every six days for ensuring the accuracy of PTR-MS. The relative humidity and temperature of inlet gas for calibration were set at 80% and 25°C respectively according to the average level (RH = 78.2% and T = 23.5°C) of ambient air in Hong Kong from 1997 to 2012 (shown in Fig. 1; data were obtained from the Hong Kong Observatory, <a href="http://www.hko.gov.hk/cis/climat\_e.htm">http://www.hko.gov.hk/cis/climat\_e.htm</a>).

# 35 2.3 DNPH-HPLC

Twenty-four hour (0:00-23:59) OVOC samples were collected by an ATEC Model 2200 automated sampler (Model 2200, Malibu, CA) once every six days in each sampling month. An ozone scrubber (Waters Corporation, Milford, MA) was used to remove ozone during carbonyl sampling with 2,4-dinitrophenylhydrazine (DNPH) cartridges (Waters Sep-Pak DNPH-silica, Milford, MA). The flow rate was regulated at 0.7 L min<sup>-1</sup> by a mass flow controller. Cartridge samples were analyzed according

to the United States Environmental Protection Agency (USEPA) Method TO-11A (USEPA, 1999). The precision of the DNPH cartridge samples was within 10% during this study.

A High Performance Liquid Chromatography (HPLC) system (Perkin Elmer Series 2000, Massachusetts 02451, USA) coupled with an ultra-violet (UV) detector operating at 360 nm was used for chemical analysis. The column for separation was a 4.6 ×

- 150 mm Hypersil ODS 5 m reversed phase column (Alltech, Deerfield, IL) at room temperature (Huang et al., 2011). The mobile phase consisted of two solvent mixtures: (A) 6:3:1 (v/v/v) water/acetonitrile/tetrahydrofuran, and (B) 4:6 (v/v) water/acetonitrile. The gradient program was 80% A/20% B for 1.5 min, followed by a linear gradient to 50% A/50% B in 8 min, and finally 100% B for 10 min. The flow rate was 2.0 mL min<sup>-1</sup> throughout the run. All solvents and water used were HPLC grade (Duksan Pure Chemicals Co., Ltd, Gyeonggi-do, Korea) and Milli-Q grade, respectively. The calibration curve
- was established by five concentration points covering the levels of interest. One calibration standard was run for every ten samples to ensure the stabilization of the instrument. Carbonyls were identified and quantified by their retention times and peak areas of the corresponding calibration standards (Cheng et al., 2014). Formaldehyde, acetaldehyde, acrolein, acetone, propionaldehyde, crotonaldehyde, 2-butanone, *i-/n*-butyraldehyde, benzladehyde, *i*-valeraldehyde, valeraldehyde, *o*tolualdehyde, *m*-tolualdehyde, *p*-tolualdehyde, hexaldehyde, and 2,5-dimethylbenzaldehydewere detected by this method. The
- detection limits for target carbonyls were below 0.45 ppbv.

# 2.4 On-line GC-FID

An on-line GC-FID analyzer (Syntech Spectras GC 955, Series 600/800, the Netherlands), which consists of two separate systems for detection of  $C_2$ - $C_5$  and  $C_6$ - $C_{10}$  hydrocarbons was used to collect VOCs speciation data continuously from May 2013 to August 2014. All the on-line GC-FID data used in this study were measured by the HKEPD

- (http://epic.epd.gov.hk/ca/uid/airdata). Data quality was assured by strict quality assurance and quality control (QA/QC) procedures. Weekly calibrations were conducted by using NPL standard gas (National Physical Laboratory, Teddington, Middlesex, UK). More details about the on-line GC-FID system can be found from previous studies in Hong Kong (Xue et al., 2014;Ou et al., 2015). The C<sub>2</sub>-C<sub>10</sub> hydrocarbon species included ethane, ethene, ethyne, propane, propene, *n*-butane, *i*-butane, 1-butene, *trans*-2-butene, *cis*-2-butene, 1,3-butadiene, *n*-pentane, *i*-pentane, 1-pentene, *trans*-2-pentene, *n*-hexane, 2-methyl
- pentane, *n*-heptane, *n*-octane, *i*-octane, benzene, toluene, ethylbenzene, *m*,*p*-xylene, *o*-xylene, 1,2,4-trimethylbenzene and 1,3,5-trimethylbenzene. Sampling resolution for the on-line GC-FID analyzer was 30 minutes, and hourly averaged data were used for further analysis. The accuracy and precision of the measurements was 20% and 10%, respectively.

#### 2.5 Off-line GC-MSD/FID/ECD

Twenty-four hour (0:00-23:59) VOC samples were collected using 2-L electropolished, conditioned stainless steel canisters and an ATEC Model 2200 automated sampler (Model 2200, Malibu, CA) once every six days in each sampling month. All canisters used in this study were pre-cleaned and evacuated by the Rowland/Blake group at the University of California, Irvine (UCI). Detailed information of the preparation and pre-conditioning of the canisters can be found elsewhere (Blake et al., 1994; Simpson et al., 2010). After sampling, canisters with air samples were sent back to the laboratory at UCI for VOCs analysis by a GC-MSD/FID/ECD system within one week of collecting the canister samples. Complete analytical details are given in

Colman *et al.* (Colman et al., 2001). Briefly, the sample flow is split into five streams, with each stream chromatographically separated on an individual column and sensed by a single detector, namely: (1) FID with a DB-1 column (60 m, I.D. 0.32 mm, film 1 mm); (2) FID with PLOT (30 m, I.D. 0.53 mm) + DB-1 (5 m, I.D. 0.53 mm, film 1 mm) columns; (3) ECD with a Restek 1701 column (60 m, I.D. 0.25 mm, film 0.50 mm); (4) ECD with DB-5 (30 m, I.D. 0.25 mm, film 1 mm) + Restek 1701 (5 m, I.D. 0.25 mm, film 0.5 mm) columns; and (5) MSD with a DB-5ms column (60 m, I.D. 0.25 mm, film 0.5 mm).

This technique was used to measure 55  $C_1$ - $C_{10}$  VOCs, including all those measured by the on-line GC-FID. The accuracy of the VOC measurements was 5%, and the measurement precision ranged from 0.5-5%.

#### **3** Results and discussion

# 3.1 Effect of ambient RH and T on PTR-MS measurement

- As described in the methodology, four-point calibrations were conducted to get the calibration curves. The experimental reaction rate coefficient (*k*) of each compound was obtained by the original input *k* value (typically 2.0  $\pm 10^{-9}$  cm<sup>3</sup> s<sup>-1</sup>) multiplied by the slope of the measured concentration to the diluted concentration of standard gas, because the volume mixing ratio (VMR) of each species is inversely related to its reaction rate coefficient (*k*) (see Eq. (1)). Experimental *k* values of selected VOC and OVOC species were obtained under different ambient conditions, and RH ranged from 25% to 100% and T ranged from 5°C
- to  $25^{\circ}$ C through this method.

$$VMR = 1.657 \times 10^{-11} \cdot \frac{U \cdot T^2 \cdot I_{RH^+} \cdot TR_{RH^+}}{k \cdot p \cdot I_{H,0^+} \cdot TR_{H,0^+}}$$
(1)

where,

VMR - the volume mixing ratio, ppbv;

U - the drift voltage, V;

T - the actual temperature in the drift tube, K.

k - the reaction rate coefficient,  $10^{-9}$  cm<sup>3</sup> s<sup>-1</sup>;

p - the drift tube pressure, mbar;

I - the numbers of detected ions, cps;

TR - the transmission factor of respective ions.

- Fig. 2 shows the effect of ambient RH on the sensitivity and experimental *k* value of HCHO and benzene. The temperature of inlet air was kept at 25°C during the series of various RH tests. A significant decrease (about 40%) of sensitivity and calibrated k value was found when the RH of inlet air increased from 25% to 100% for HCHO (m/z 31). However, the sensitivity and experimental *k* value of benzene (m/z 79) did not vary much while the RH of inlet air changed with a variance of less than 5%. The variation patterns of sensitivity and experimental *k* value for the other species were similar to benzene. Therefore, only
- benzene is plotted to represent other species in the gas standard. HCHO was found to have lower sensitivity (less than 2.0 ncps ppbv<sup>-1</sup>) than benzene (about 20 ncps ppbv<sup>-1</sup>), since the backward reaction of protonated formaldehyde with H<sub>2</sub>O is not negligible, and the ratio between the forward and backward reaction was determined to be about 6000 by Spanel and Smith (2008). The effect of ambient temperature on sensitivity and experimental *k* value of HCHO and benzene is plotted in Fig. 3 under different RH. Both the sensitivity and experimental *k* value significantly dropped by 30 40% when the temperature of
- inlet air increased from 5°C to 25°C. Both sensitivity and experimental *k* value maintained a relatively stable level (variance less than 5%) for benzene when the temperature of inlet air was changed under the given RH, and the same for the other species in the standard gas excluding HCHO. A quadratic polynomial fit was found to be the best fit model ( $r^2 > 0.98$ ) for the experimental *k* value of HCHO with a single variable (RH or T of inlet air). Because the RH and T of inlet gas for calibration were set at 80% and 25°C respectively, R<sub>(RH,T)</sub> (the ratio of *k* value under given ambient conditions to the experimental *k* value
- under 80% and 25°C) was chosen to explore the integrated effects of ambient RH and T on experimental k value. Poly2D regression, which used a binary quadratic equation, was found to be the best nonlinear surface simulation of R<sub>(RH,T)</sub>, ambient RH and T for HCHO in this study, and the following equation (Eq. (2)) shows the relationship between them.

 $\mathbf{R}_{(\mathrm{RH},\mathrm{T})} = \mathbf{r}_0 + \mathbf{a} \ \mathbf{R}\mathbf{H} + \mathbf{b} \ \mathbf{T} + \mathbf{c} \ \mathbf{R}\mathbf{H}^2 + \mathbf{d} \ \mathbf{T}^2 + \mathbf{f} \ \mathbf{R}\mathbf{H} \ \mathbf{T}$ (2)

where,

 $R_{(RH,T)}$  - the ratio of experimental reaction rate under given RH & T to calibrated reaction rate under 80% and 25°C; RH - the relative humidity of inlet gas/air;

T - the temperature of inlet gas/air.

- The constant term  $r_0$  and coefficients a, b, c, d, and f, as well as the correlation coefficients of all species in the gas standard are listed in Table 1. The difference of  $R_{(RH,T)}$  was within 10% for all species except HCHO (m/z 31) when the condition of inlet air changed within controlled range for RH (25% ~ 100%) and T (5°C ~ 25°C). The  $R_{(RH,T)}$  for formaldehyde differs significantly because of humidity effects on HCHO measurement by the PTR-MS. Excellent correlation was found between  $R_{(RH,T)}$ , ambient RH and T for HCHO (m/z 31) with a correlation coefficient of 0.97. The relationship between these three
- factors is plotted in Fig. 4.

The relationship of experimental *k* value and absolute humidity (AH) in sample air for HCHO is plotted in Fig. 5. The best simulation (r = 0.92) was found following the function:  $k_{(AH)} = 0.0007 \text{ AH}^2 - 0.0568 \text{ AH} + 1.8139$ , where  $k_{(AH)}$  stands for the experimental *k* value of HCHO, AH stands for the absolute humidity in sample air. Fig. 6 shows the comparison between concentrations of HCHO measured during the campaign at MK after correction by  $R_{(RH,T)}$  and  $k_{(AH)}$  separately. Good agreement

during the field sampling for HCHO was found between the above two correction methods with a slope of 1.02 and a correlation coefficient of 0.99 (as shown in Fig. 6).

A relationship between the normalized detection sensitivity, S (ncps ppbv<sup>-1</sup>), of HCHO and its humidity dependence at a certain E/N condition was raised by Inomata et al. (2008) as follows:

$$S = \frac{m}{n} \cdot \frac{a}{[H_2O]_{sample} + b} \cdot \frac{1}{88.0}$$
(3)

where,

m - the detection sensitivity of HCHO by direct introduction, ncps ppbv-1;

n - the detection sensitivity of HCHO by Dynamic dilution, ncps ppbv<sup>-1</sup>;

a, b – fitting parameters obtained from the simulation equation y = a / (x + b) for the signal intensity of HCHO,

 $[H_2O]_{sample}$  - the water vapor concentration in sample air, mmol mol<sup>-1</sup>.

- Ion counts are divided by 88.0 because the humidity dependence was measured with HCHO of 88.0 ppbv in the dynamic dilution method in this study. The relationship of the signal intensity of HCHO and the water vapor concentration in sample air is plotted in Fig. 7. Simulated fitting parameters, a and b, are 5404  $\pm$  72 (ncps mmol mol<sup>-1</sup>) and 26  $\pm$  0.4 (mmol mol<sup>-1</sup>), respectively. The value of m/n is 1.09 in this study, and it is about 1.2 times of the (k<sub>a</sub>/k<sub>-a</sub>)/a value. This result agrees with the study of Inomata et al. (2008). Hence, the normalized detection sensitivity of HCHO in this study can be determined as a
- function of the water vapor concentration in sample air as Eq. (4).

$$S = \frac{67.18}{AH_{sample} + 26.04}$$
(4)

Fig.8 shows the comparison of the measured detection sensitivity of HCHO and the normalized detection sensitivity obtained from Eq. (4). Good agreement was found between the measured detection sensitivity and the normalized detection sensitivity of HCHO with a slope of  $1.18 \pm 0.01$  (r = 0.998). Since the normalized detection sensitivity of HCHO was 1.18 times of the

35 measured detection sensitivity, lower HCHO concentration will be obtained by using the normalized detection sensitivity with a percentage of about 15.3%.

# 3.2 PTR-MS versus DNPH-HPLC

Daily averaged PTR-MS data were used to compare the ambient OVOC measurements by PTR-MS and by DNPH-HPLC. The sum of acetone and propionaldehyde measured by DNPH-HPLC was used to compare with PTR-MS, because acetone and propionaldehyde were both detected by PTR-MS at m/z 59 (de Gouw and Warneke, 2007). As shown in Fig. 9, good linear

- correlations were found for formaldehyde, acetaldehyde and MEK with correlation coefficients of 0.79, 0.75, 0.93, respectively, and with slopes of 1.00 ±0.10, 1.10 ±0.33, and 0.88 ±0.058, respectively. Concentrations of HCHO measured by the PTR-MS in Fig. 9B were the corrected data by the Poly 2D regression of RH and T. For acetone, the slope is 0.76 ±0.23 (PTR-MS to HPLC) with a correlation coefficient of 0.60, and its offset of 1.88 could be caused by lower detection limit of PTR-MS for acetone (about 30 pptv) than that of DNPH-HPLC (260 pptv), and some other interferences on m/z 59.
- Concentrations of HCHO directly measured by the PTR-MS, corrected by Poly 2D regression of ambient RH and T, and corrected by absolute humidity in sample air are plotted in Fig 9 for comparison. Since the conditions of inlet air for calibration were set at RH = 80% and T = 25°C, which were close to the average level of ambient air in Hong Kong from 1997 to 2012 (shown in Fig. 2), uncorrected concentrations of HCHO measured by PTR-MS were only slightly lower than that by DNPH-HPLC with a percentage of 8.0% (r = 0.68). Corrected concentrations of HCHO by absolute humidity in sample air also
- correlated well with that by DNPH-HPLC (slope = 0.91, r = 0.77), because of the excellent correlation between these two different correction methods discussed above. However, correction for HCHO concentrations by Poly 2D regression of RH and T which has a better slope than the others is a better choice. Moreover, it is easy to measure RH and T of ambient air for further correction of HCHO data by PTR-MS in practice.

#### 3.3 PTR-MS versus on-line GC-FID

- Hourly data were used for conducting inter-comparison between PTR-MS and on-line GC-FID. Because the ion signal at m/z 107 reflects C<sub>2</sub>-benzenes (the sum of *p*-xylene, *m*-xylene, *o*-xylene, ethylbenzene and benzaldehyde) (de Gouw and Warneke, 2007), C<sub>2</sub>-benzens measured by on-line GC-FID were compared with the ion signal at m/z 107 of PTR-MS. Time series concentrations of benzene, toluene, and C<sub>2</sub>-benzenes measured by PTR-MS and on-line GC-FID are plotted in Fig. 10. Excellent correlation was found for toluene with a slope of 1.01 ±0.01 and a correlation coefficient of 0.96. Good linearities
- were observed for benzene and C<sub>2</sub>-benzenes between PTR-MS and online GC-FID, with correlation coefficients of 0.92 and 0.96, respectively. However, the concentration of benzene detected by PTR-MS was 65% higher than that by on-line GC-FID, and the concentration of C<sub>2</sub>-benzenes detected by PTR-MS was 17% lower than that by on-line GC-FID. The significant difference of benzene detected by PTR-MS and on-line GC-FID could mainly be due to the fragments from ethylbenzene and propylbenzene to m/z 79. Rogers et al. (2006) raised an equation (Eq. (5)) to calculate the contributions from ethylbenzene
- and propylbenzene to m/z 79.

$$[Benzene] = [M79] - \frac{S_{Ethylbenzene}}{S_{Benzene}} BF_{Ethylbenzene} [Ethylbenzene] - \frac{S_{Propylbenzene}}{S_{Benzene}} BF_{Propylbenzene[Propylbenzene]}$$
(5)

where,

[Benzene] - volume mixing ratio of benzene;

[M79] - volume mixing ratio of m/z 79;

[Ethylbenzene] - actual volume mixing ratio of ethylbenzene;

[Propylbenzene] - actual volume mixing ratio of propylbenzene;

 $S_{\text{x}}\!/S_{\text{Benzene}}-\text{ratio of the ionization efficiencies};$ 

BF - fraction of m/z 79 ion product that each produces upon ionization.

Since propylbenzene was not measured in this study, only ethylbenzene was used to correct the concentrations of benzene and  $C_2$ -benzenes measured by PTR-MS following Eq. (6) and Eq. (7):

$$[Benzene] = [M79] - 0.2235 [M107]$$
(6)  
$$[C_2-benzenes] = 1.2235 [M107]$$
(7)

5 where,

[C<sub>2</sub>-benzenes] - volume mixing ratio of C2-benzenes;

[M107] - volume mixing ratio of m/z 107;

Fig. 10 (A and C) shows the correlation between corrected concentrations of benzene and C<sub>2</sub>-benzenes for PTR-MS and their measured concentrations by on-line GC-FID at MK AQMS. The agreement between the two different sampling techniques for benzene is better after correction by Eq. (3) with a slope of  $1.23 \pm 0.01$  and the correlation coefficient of 0.95. Excellent agreement (slope =  $1.02 \pm 0.01$ , r = 0.96) between the two different sampling techniques for C<sub>2</sub>-benzenes is achieved using the corrected concentrations of C<sub>2</sub>-benzenes by PTR-MS. Since toluene to benzene ratio is one of the key indicators of

determining VOC sources and C<sub>2</sub>-benzenes are related to vehicular emissions and solvent usage, accurate measurements are required in studies on source apportionment of ambient VOCs. Above two equations (Eq. (4) and (5)) offer a good correction

method for the concentrations of benzene and C2-benzenes measured by PTR-MS in urban roadside area of Hong Kong.

## 3.4 PTR-MS versus off-line GC-MSD/FID/ECD

The inter-comparison between the PTR-MS data averaged over the 24-hour canister sampling interval and off-line GC-MSD/FID/ECD analysis of the canisters are displayed in Fig. 11. Corrected concentrations of benzene and  $C_2$ -benzenes by the PTR-MS were used for comparison. Acceptable linear regressions were found for benzene, toluene and  $C_2$ -benzenes between

- the PTR-MS and the off-line GC-MSD/FID/ECD with correlation coefficients of 0.62, 0.96 and 0.92, respectively. However, the mixing ratios of toluene and C<sub>2</sub>-benzenes measured by PTR-MS were lower than that measured by off-line canister samples with percentages of 22%, 33%, respectively. The maximum offsets from the 1:1 line were usually found at high concentrations, which could be mainly caused by the different sampling time resolution for each sample of the PTR-MS and canister samples. The canister samples were 24-hour samples with constant flow, while PTR-MS detected a total group of 17 different species
- with a 30 seconds cycle. From previous studies on comparison between the PTR-MS and the GC-MSD/FID/ECD, the slope for toluene ranged 0.52 ~ 1.18, and the slope for C<sub>2</sub>-benzenes ranged 0.58 ~ 3.20 (Kato et al., 2004; Christian et al., 2004; de Gouw et al., 2003; Warneke et al., 2001), so PTR-MS and off-line GC-MSD/FID/ECD analysis were comparable for aromatic hydrocarbons in this study.

#### 3.5 Comparison with other studies

Inter-comparisons between PTR-MS and other alternative technologies are summarised in Table 2. Most studies were conducted in urban areas, suburban areas, coastal areas, forested areas, or the free troposphere. Overall, most of the slopes of PTR-MS to alternative methods agreed to within less than the PTR-MS measurement accuracy of 20%.

Few studies have conducted inter-comparisons between PTR-MS and alternative methods for OVOCs (e.g. HCHO, acetaldehyde, acetone and MEK). Even fewer studies have investigated inter-comparison between PTR-MS and DNPH-HPLC.

- In most studies, a DOAS instrument and Hantzsch monitor were usually used to measure ambient HCHO. Good agreement was found between PTR-MS and DOAS in both mountain areas (slope = 1.01) (Inomata et al., 2008) and urban areas (slope = 1.17) (Warneke et al., 2011b). HCHO measurement by DNPH-HPLC was compared to PTR-MS in an atmosphere simulation chamber by A. Wisthaler et al. (2008), and significant lower HCHO levels by DNPH-HPLC was observed in dry synthetic air because of hydrazine-to-hydrazone conversion. PIT-MS, AP-CIMS, GC-MS/FID were chosen as alternative monitoring
- methods for acetaldehyde, acetone and MEK. Relatively higher acetaldehyde levels by PTR-MS was found while PTR-MS

versus PIT-MS (Warneke et al., 2011a) and PTR-MS versus on-line GC-MS (de Gouw et al., 2003) with slopes of 1.25 and 1.56, respectively. Better agreement was obtained between PTR-MS and DNPH-HPLC for acetaldehyde (slope = 1.10, r = 0.75) in this study. PTR-MS agreed well with AP-CIMS, online GC-MS and on-line GC-MS-FID for acetone, with slopes ranging from 1.00 to 1.18. In this study, however, the slope is 0.76 (PTR-MS to HPLC) with the offset of 1.88, which could

- have resulted from other interferences on m/z 59. The slopes of MEK varied in a large ranger (0.85 2.51) in previous studies, but a reasonable correlation between PTR-MS and DNPH-HPLC for MEK (slope = 0.88, r = 0.93) was found in this study. Inter-comparisons between PTR-MS and alternative methods (both on-line and off-line methods) for benzene, toluene and C<sub>2</sub>benzenes have been conducted by previous studies in various areas. For benzene, PTR-MS was always reasonably comparable to other on-line or off-line technologies, with slopes ranging from 0.82 to 1.22. In this study, PTR-MS correlated well with
- off-line canister measurements for benzene (slope = 1.05), but a higher slope of 1.23 was found between PTR-MS and on-line GC-FID, which could be due to interferences (e.g. propylbenzene) at m/z 79. Lower toluene levels measured by PTR-MS as compared to off-line canister measurements was found in this study, with a slope of 0.78 (r = 0.96). Lower toluene levels measured by PTR-MS was also reported between PTR-MS versus DOAS (Jobson et al., 2010) and PTR-MS versus on-line GC-FID (Kato et al., 2004) in an urban area. Similarly, lower C<sub>2</sub>-benzenes levels measured by PTR-MS to off-line canister
- measurements was found with a slope of 0.67 (r = 0.92) in this study, which could have resulted from the much longer sampling resolution of off-line canister sample. The slopes of C<sub>2</sub>-benzenes varied from 0.59 to 3.20 in previous studies with different area types.

#### 4 Conclusions

The effect of ambient RH and T on HCHO measurements by PTR-MS was further investigated in this study. Due to the 20 backward reaction of protonated HCHO with H<sub>2</sub>O, the sensitivity of HCHO decreased significantly when ambient RH and/or T varied. Meanwhile, calibrated reaction rate coefficient of HCHO and H<sub>3</sub>O<sup>+</sup> also decreased significantly. The combined effect of RH and T on HCHO measurement by PTR-MS was explored in this study. A Poly 2D regression was found to be the best nonlinear surface simulation of R<sub>(RH,T)</sub>, ambient RH and T for HCHO, following the equation R<sub>(RH,T)</sub> = 1.63 - 3.81  $10^{-3}$  RH +  $1.92 \cdot 10^{-2}$  T + 5.10  $10^{-5}$  RH<sup>2</sup> - 6.41  $10^{-4}$  T<sup>2</sup> - 3.76  $10^{-4}$  RH T. Through a filed sampling study at an urban roadside area at MK

AQMS, correction of HCHO concentrations by both ambient RH and T and by absolute humidity in sample air agreed well with each other (slope = 1.02, r = 0.99).
 A field study of OVOCs and VOCs using the PTR-MS was conducted at MK in Hong Kong from May 2013 to August 2014.

Good agreement was found between PTR-MS and DNPH-HPLC for formaldehyde (slope = 1.00, r = 0.79), acetaldehyde (slope = 1.10, r = 0.75) and MEK (slope = 0.88, r = 0.93). For acetone, its offset of 1.88 could be resulted from lower detection

- 30 limit of PTR-MS for acetone than that of DNPH-HPLC, and some other interferences on m/z 59. Moreover, correction for HCHO concentrations by Poly 2D regression of ambient RH and T was found to be better than directly correcting by absolute humidity in the sample air. Aromatic hydrocarbons measurements by PTR-MS were inter-compared with on-line GC-FID and off-line canister measurements using GC-MSD/FID/ECD. After correcting benzene and C<sub>2</sub>-benzenes levels that were measured by the PTR-MS, which could be affected by fragments from ethylbenzene and propylbenzene at m/z 79, good
- agreements were found between PTR-MS and on-line GC-FID for toluene (slope = 1.01, r = 0.96) and C<sub>2</sub>-benzenes (slope = 1.02, r = 0.96), but higher benzene level was still detected by the PTR-MS when compared with that by on-line GC-FID (slope = 1.23, r = 0.95). For the inter-comparisons between PTR-MS and off-canister measurements using GC-MSD/FID/ECD, benzene showed good agreement with a slope of 1.05 (r = 0.62), underestimated toluene and C<sub>2</sub>-benzenes levels by PTR-MS were obtained with slopes of 0.78 (r = 0.96) and 0.67 (r = 0.92), respectively. In summary, the PTR-MS instrument can be