# Peer review of "Measuring OVOCs and VOCs by PTR-MS in an urban roadside microenvironment of Hong Kong: relative humidity and temperature dependence, and field inter-comparisons"

_Atmospheric Measurement Techniques, 2016_

## Referee Comment (RC1) · Anonymous Referee #1 · 14 Jul 2016

Report on AMT-2016-130 The paper reports on a field inter-comparison between PTR-MS and several other techniques in the detection of VOCs and OVOCs at an urban site. The paper is well-structured and provides a thorough overview of the current state of knowledge, as well as sufficient details of the analytical approach of the study. The data treatment and discussion on the whole are comprehensive and sound. The presented figures and tables are clear and offer an excellent overview of the data obtained. In discussing and comparing the data between PTR-MS and the other techniques, however, one aspect that is missing is the recognition of the high time resolution of measurements by PTR-MS compared to the other techniques; it all very well to compare

absolute concentrations of different species detected by each technique in comparison to PTR-MS, but the rapid and continuous analysis by PTR-MS are somewhat downplayed, yet are certainly a key feature of the system that make it particularly suited to long-term VOC monitoring. The discussion on the observed discrepancies could also be expanded upon. At present most of the emphasis is on the humidity-dependent performance of the PTR-MS detection, yet not much is said about potential under- or overestimation of the data by the other techniques with which PTR-MS is compared. Further specific comments are as follows: Page 2, lines 32-34: Please acknowledge the first reports of these reactions in the detection of HCHO by PTR-MS, namely: Hansel et al. Int. J. Mass Spectrom., 167/168, 697–703, 1997. Page 3, line 10: which model PTR-MS was used? Page 3, line 13: H3O+ was used as the "reagent ion", not the "ion source". Please correct. Page 3, line 20: replace "flow tube" with "inlet line" to avoid confusion with the flow drift tube of the instrument (reaction chamber) Page 3, line 30: how were the accuracy and precision values stated for the PTR-MS instrument determined? The same applies to these values presented for the other instruments in the study. It would also be desirable if the authors presented the limits of detection of the VOCs and OVOCs presented in the inter-comparison, perhaps most suitably in the form of a table. Page 5, Eq. 1: this is an unusual presentation of how to calculate the VMR from the analyte and reagent ion signal intensities in PTR-MS. If the authors choose to present it like this, I think that further details are needed of how they arrived at this arrangement, either by explanation or by a suitable reference. Please also indicate how the value of the constant was reached. Page 5, lines 28-30, discussion relating to sensitivity dependence of HCHO to relative humidity: the authors should acknowledge and discuss similar work performed on the same VOC standard using the same equipment, in which similar observations were made, namely: Beauchamp et al. Meas. Sci. Technol., 24, 125003, 2013. How do the present measurements compare to those reported in the aforementioned article? Page 9, lines 3-4: the authors start this section by referring to a comparison between PTR-MS and DNPH-HPLC data for acetone and propionaldehyde, but in the next sentence discuss other compounds and

an unrelated figure. The discussion on the former reappear at the end of this paragraph but with no presentation of the data: are the data for acetone not shown? Consider repositioning the discussion on those data to the start of the paragraph and indicate that the data are not presented. Figures 9 and 11: where are the error bars for the DNPH-HPLC and off-line GC-MSD/FID/ECD analyses, or is there no measurement error associated with these systems? Figure 11 and 12 captions: the authors should elaborate in the caption on what the "corrected" data are. Errors.

Throughout: Please be consistent in the use of VOC and VOCs for singular and plural, respectively (similarly for OVOC). Abstract, line 24 and p2, line 12: place parentheses around ECD and not electron capture detection. Page 2, line 29: change to "HCHO can be protonated by the following reaction". Page 2, line 32: change "is just slight higher" to "is just slightly higher". Page 7, line 22: "C2-benzenes" not "C2-benzens". Page 9, line 24: "field" not "filed" sampling study. Figure 1 caption: should be "dashed lines" not "das lines".

---

## Referee Comment (RC2) · Anonymous Referee #2 · 19 Jul 2016

Cui et al. show VOC measurements from PTR-MS in Hong Kong. They also discuss humidity dependence of formaldehyde in PTR-MS and inter-comparison with several other techniques, including DNPH, canister samples, online GC-FID. However, this manuscript generally does not provide much new information and new technique. Many valuable experiences from over 20 years of work in the PTR-MS community are not fully reflected in the data processing procedures in this manuscript. The inter-comparison results are not as good as those previously reported in the literature, but the authors do not provide good reasons about it. Thus, I do not think this manuscript is suitable to publish in AMT, unless this manuscript is totally re-written and provide more information

that can support new idea/technique.

Specific comments

1. Hong Kong is just one city in Pearl River Delta region. Thus, the authors might want to introduce more previous work in PRD region, rather than just Hong Kong. Several researchers have reported PTR-MS results in PRD region [Wang et al., 2016], and more in other parts of China, but the authors do not acknowledge these references in the introduction.

2. Formaldehyde humidity dependence in PTR-MS. Several papers have discussed this issue previously [Inomata et al., 2008; Vlasenko et al., 2010; Warneke et al., 2011]. Especially, Valsenko et al. derived explicit equations to fit the sensitivity vs. absolute humidity, based on the equilibrium between forward and backward reactions. In this manuscript, the authors tried to fit the observed rate coefficient (k) and sensitivity with relative humidity, temperature and absolute humidity. All of these tests are just using one dataset with different parameters/equations to derive curves for the correction, but without knowing the physical meaning of parameters/equations. The best way to do it should use the equations shown in Valsenko et al. Thus, Figure 2-8 should be replaced with one Figure similar to Fig. 5 in Valsenko et al.

3. PTR-MS operations: how often the instrument is calibrated, how is background determined, how often do you do background, how often the humidity dependence curve for formaldehyde is conducted (just one in 2 years?).

4. PTR-MS data processing: how signal is normalized, do you see sensitivity drift with time, how background is interpolated (especially formaldehyde). The authors should apply the widely accepted data reduction methods shown in many previous papers (most important ones [de Gouw and Warneke, 2007; de Gouw et al., 2003]).

5. Inter-comparison: In addition to PTR-MS, the QA/QC procedures should be provided to evaluation the data quality. Previous studies also found many limita-

tions/interferences for DNPH method, such as formaldehyde [Wisthaler et al., 2008], aldehydes [Herrington and Hays, 2012] and ketones [Ho et al., 2014]. Without acknowledging these interferences, the inter-comparison is meaningless. From Figure 9 in the manuscript, the bad agreement for acetone might be a DNPH issue.

6. Eq. 7: If you use m107 to calibrate C2 benzenes, then C2 benzenes should not be corrected. Where is 0.2235 from? Note that only ethylbenzene fragments significantly. Thus, Eq. 6 should be [Benzene]=[m79]-0.2235*[m107]*f. f is the fraction of ethylbenzene in ethylbenzene+xylenes, which can be from GC-FID measurements. Then, PTR-MS measurements would somewhat rely on GC-FID measurements. How would this reliance affect inter-comparison?

7. Figure 1 should only contain the periods with only PTR-MS measurements. The time series of VOCs should be also shown.

References:

de Gouw, J., and C. Warneke (2007), Measurements of volatile organic compounds in the earth's atmosphere using proton-transfer-reaction mass spectrometry, Mass Spectrometry Reviews, 26(2), 223-257.

de Gouw, J. A., P. D. Goldan, C. Warneke, W. C. Kuster, J. M. Roberts, M. Marchewka, S. B. Bertman, A. A. P. Pszenny, and W. C. Keene (2003), Validation of proton transfer reaction-mass spectrometry (PTR-MS) measurements of gas-phase organic compounds in the atmosphere during the New England Air Quality Study (NEAQS) in 2002, Journal of Geophysical Research-Atmospheres, 108(NO. D21, 4682), doi:10.1029/2003JD003863.

Herrington, J. S., and M. D. Hays (2012), Concerns regarding 24-h sampling for formaldehyde, acetaldehyde, and acrolein using 2,4-dinitrophenylhydrazine (DNPH)-coated solid sorbents, Atmospheric Environment, 55(0), 179-184.

Ho, S. S. H., J. C. Chow, J. G. Watson, H. S. S. Ip, K. F. Ho, W. T. Dai, and J. Cao

(2014), Biases in ketone measurements using DNPH-coated solid sorbent cartridges, Analytical Methods, 6(4), 967-974.

Inomata, S., H. Tanimoto, S. Kameyama, U. Tsunogai, H. Irie, Y. Kanaya, and Z. Wang (2008), Determination of formaldehyde mixing ratios in air with PTR-MS: laboratory experiments and field measurements, Atmospheric Chemistry and Physics, 8(2), 273-284.

Vlasenko, A., A. M. Macdonald, S. J. Sjostedt, and J. P. D. Abbatt (2010), Formaldehyde measurements by Proton transfer reaction – Mass Spectrometry (PTR-MS): correction for humidity effects, Atmos. Meas. Tech., 3(4), 1055-1062.

Wang, B., et al. (2016), The contributions of biomass burning to primary and secondary organics: A case study in Pearl River Delta (PRD), China, Science of The Total Environment, 569–570, 548-556.

Warneke, C., et al. (2011), Airborne formaldehyde measurements using PTR-MS: calibration, humidity dependence, inter-comparison and initial results, Atmos. Meas. Tech., 4(10), 2345-2358. Wisthaler, A., et al. (2008), Technical Note: Intercomparison of formaldehyde measurements at the atmosphere simulation chamber SAPHIR, Atmospheric Chemistry and Physics, 8(8), 2189-2200.
* * *

---

## Referee Comment (RC3) · Anonymous Referee #3 · 21 Jul 2016

Review. Measuring OVOCs and VOCs by PTR-MS in an urban roadside microenvironment of Hong Kong: relative humidity and temperature dependence, and field inter-comparisons by Long Cui et al.

The authors have applied a suite of techniques to measure VOC and OVOC at a roadside site in Hong Kong and have compared the data to those generated by a PTR-MS. Measurements of formaldehyde are examined in detail and a method has been developed to address the RH and T dependence of the calibration for future studies. This approach seems reasonable. OVOCs measured by PTR-MS are compared with

a DNPH method and aromatics measured by PTR-MS are compared to in-situ GC-FID and canister samples. Besides my question regarding the fitting method below, the comparison has been done well enough, very much along the line of previous groups.

However, I cannot help feeling though that there is an opportunity missed here. Surely the GC-FID also measured the OVOC (i.e. not just the aromatics)? Can their peaks not be calibrated by carbon number and compared with the PTR-MS/DNPH/can methods as well? The result would be an assessment/validation of the in-situ GC-FID measurement of OVOC which could be of use to the group in future measurement campaigns where the PTR-MS is not available. Moreover, do the canister data not deliver values of acetone, propanal and acetaldehyde for comparison? This again would be an interesting comparison extension to this study. There has been much discussion of potential canister artifacts previously but they may compare well under the conditions where ozone is low.

One of the greatest problems in sampling and measuring OVOC will be ozone as it can make OVOC (and remove reactive alkenes) when trapped in canisters or concentrators. If ozone was measured at this site is would be very interesting to compare the degree of fit with ozone levels. Most city roadside sites show low ozone, due to the titration of NO, but in the afternoon it is likely that ozone levels will increase as photochemical production and down-mixing kick in. This would mean correlation could be expected to deteriorate later in the afternoon if ozone reactions become important. Examining this influence of local ozone of the quality of fit would be a very interesting addition to this paper. It is likely to impact the GC and DNPH methods more than the PTR-MS.

For the correlation plots a simple y=mx+c form is used. This assumes that the x-axis parameter is correct and without error. More appropriate in this case would be to use orthogonal distance regression to account for error in both axes, since the DNPH method will also contain errors to some degree.

The PTR-MS accuracy is given as 20% and the precision as "about 10%". Since this

paper is an instrumental comparison I think it is necessary to expand on this and explain where the 20% comes from and the measurement precision of each species. Likewise the GC-FID accuracy and precision is given as the same as for the PTR-MS, but without explanation. How these figures arrived at should be given in more detail.

The potential interference of ozone may be an explanation of the relatively poor fit of PTR-MS vs DNPH for acetaldehyde (and also acetone) in figure 9 (D and E). It might be illuminating to color the points as a function of daily average ozone (if available).

Minor points. Introduction, line 4. Perhaps a more relevant reference concerning the human health impacts would be Lelieveld et al. (doi:10.1038/nature15371).

"Special attention has been paid on the characteristics" should be "paid to"

Section 2.2, give details of the particulate filter (i.e. material, pore size, how often changed).

Section 2.4 line 20. What were these strict QA/QC procedures? If necessary give the reference where they are detailed directly afterwards.

Section 3.3. line 2, was glyoxal measured, perhaps it can contribute to mass 59?

Section 3.3 line 3 "benzens" should be benzenes

Figure 8 Xaxis label is misspelt "Measuremeasured"

Conclusions, line 24, filed should be field.

---

## Author Comment (AC1) · 24 Aug 2016

The paper reports on a field inter-comparison between PTRMS and several other techniques in the detection of VOCs and OVOCs at an urban site. The paper is well-structured and provides a thorough overview of the current state of knowledge, as well as sufficient details of the analytical approach of the study. The data treatment and discussion on the whole are comprehensive and sound. The presented figures and tables are clear and offer an excellent overview of the data obtained.

Response: we thank the reviewer very much for the valuable comments and suggestions, which can help to improve the manuscript substantially. In the revised manuscript, we have addressed these comments, and adopted the suggestion to synthesize our results and compare against existing findings of previous studies. For clarity, the reviewer's comments are listed below in black italics, while our responses and changes in manuscript are shown in blue and red, respectively.

In discussing and comparing the data between PTR-MS and the other techniques, however, one aspect that is missing is the recognition of the high time resolution of measurements by PTR-MS compared to the other techniques; it all very well to compare absolute concentrations of different species detected by each technique in comparison to PTR-MS, but the rapid and continuous analysis by PTR-MS are somewhat downplayed, yet are certainly a key feature of the system that make it particularly suited to long-term VOC monitoring.

Response: thanks for the helpful comments and suggestions. Because each cartridge and canister sample was collected during 24 hours. It is not appropriate to plot the time series of PTR-MS v.s. DNPH-HPLC (or off-line canister). But the time series of benzene, toluene, C2-benzenes obtained by PTR-MS and on-line GC-FID were plotted

in Fig. 10 as in the revised manuscript.

"Time series of benzene, toluene and C2-benzenes results obtained by on-line GC-FID and PTR-MS during the sampling period were plotted in Fig. 10."

[Figure]

"Figure 10. Measurement results for benzene, toluene and C2-benzenes obtained during the field study at MK in Hong Kong. The black lines show the on-line GC-FID data, and the green lines show the PTR-MS results."

The discussion on the observed discrepancies could also be expanded upon. At present most of the emphasis is on the humidity-dependent performance of the PTR-MS detection, yet not much is said about potential under- or overestimation of the data by the other techniques with which PTR-MS is compared.

Response: thanks for the excellent comments and suggestions. We have expand the mentions part in the revised manuscript as follows.

"Beauchamp et al. (2013) used a similar mixed gas standard for PTR-MS calibration, low sensitivity of HCHO was also found below 2 ncps ppbv-1 when the RH ranged from 20% to 100%. Moreover, strong nonlinear dependence on RH for HCHO was found by Beauchamp et al. (2013), and the sensitivity of HCHO significantly decreased by about 50% when the RH of inlet air increased from 20% to 100%, which is

comparable to our study."

"Wisthaler et al. (2008) reported the inter-comparison between PTR-MS and DNPH-HPLC in an atmosphere simulation chamber, good agreement was found between PTR-MS and DNPH-HPLC while ambient air was introduced into the chamber, but the concentration of HCHO measured by DNPH-HPLC was less than by PTR-MS, which could be caused by some interferences for DNPH-HPLC method or the varying performance of the KI ozone scrubber. Overestimation of DNPH-HPLC for HCHO in the presence of $NO_2$ was also reported by Herrington and Hays (2012), because NO can be oxidized to $NO_2$ in the upstream ozone scrubber, and $NO_2$ will react with DNPH to form 2,4-dinitrophenylazide (DNPA), which has the similar chromatographic properties with the formaldehyde-DNP-hydrazone. Hence, the intercept of -0.03 for HCHO inter-comparison between PTR-MS and DNPH-HPLC in this study may be explained by the interference of $NO_2$ because of the high $NO_x$ levels at the roadside sampling site."

"Low acetaldehyde collection efficiencies (CEs), ranging from 1 to 62% was found by Herrington et al. (2007) for the typical 24-hour sampling period which can lead to the underestimation of acetaldehyde by DNPH-HPLC method. And this artifact is consistent with the result for acetaldehyde inter-comparisons in this study. It was found that ketone concentrations determined by DNPH-HPLC method could be underestimated by 35 ~ 80% under high RH (>50%) condition when the temperature is about 22°C (Ho et al., 2014). This DNPH issue could explain the 12% difference between PTR-MS and DNPH-HPLC for MEK and the relative bad agreement for acetone in our study."

Further specific comments are as follows: Page 2, lines 32-34: Please acknowledge the first reports of these reactions in the detection of HCHO by PTR-MS, namely: Hansel et al. Int. J. Mass Spectrom., 167/168, 697–703, 1997.

Response: thanks for the comments. The reference was added in the revised manuscript.

Page 3, line 10: which model PTR-MS was used?

Response: thanks for the comments. Model PTR-QMS 500 was used in this study. It was added in the revised manuscript as follows.

"A commercially available PTR-MS instrument (PTR-QMS 500, IONICON Analytik GmbH, Innsbruck, Austria) was used in this study. PTR-MS instrument has been described in detail elsewhere (Lindinger et al., 1998;de Gouw et al., 2003;de Gouw and Warneke, 2007)."

Page 3, line 13: H3O+ was used as the "reagent ion", not the "ion source". Please correct.

Response: thanks for the comments. It has been revised in the manuscript.

Page 3, line 20: replace "flow tube" with "inlet line" to avoid confusion with the flow drift tube of the instrument (reaction chamber).

Response: thanks for the comments. It has been revised in the manuscript.

Page 3, line 30: how were the accuracy and precision values stated for the PTR-MS instrument determined? The same applies to these values presented for the other instruments in the study. It would also be desirable if the authors presented the limits of detection of the VOCs and OVOCs presented in the inter-comparison, perhaps most suitably in the form of a table.

Response: thanks for the useful comments and suggestions. We agree the point of the reviewer, and we expand on the methodology part and give more explanation in the revised manuscript. Several parameters (reaction rate coefficient, fragmentation, flow rate, gas standard…) lead to accuracy, most important is the reaction rate coefficient (Salisbury et al., 2003). One important part of our study was to study the experimental reaction rate coefficient, and it also point out the significance of our study. The related information were added in the manuscript as follows.

"The accuracy and the measurement precision of the PTR-MS was 20% and 10%, respectively. The accuracy of PTR-MS measurement was dependent on the accuracy of the reaction rate coefficient, fragmentation, gas standard and flow rate (Salisbury et al.,

2003;Kim et al., 2009). The precision was determined based on the standard deviation of the background signal at each mass during 5-min average measurement for each specie."

"The accuracy was based on weekly span checks and monthly calibration. The precision was based on the 95% probability limits for the integrated precision check results (Ling et al., 2013;Lyu et al., 2016)."

Page 5, Eq. 1: this is an unusual presentation of how to calculate the VMR from the analyte and reagent ion signal intensities in PTR-MS. If the authors choose to present it like this, I think that further details are needed of how they arrived at this arrangement, either by explanation or by a suitable reference. Please also indicate how the value of the constant was reached.

Response: thanks for the good comments and suggestions. We have changed the equation to a more used equation, which can be found in the reference (de Gouw and Warneke, 2007). The manuscript was revised as follow.

"Experimental $k$ values of selected VOC and OVOC species were obtained under different ambient conditions, and RH ranged from 25% to 100% and T ranged from 5$^\circ$C to 25$^\circ$C through this method.

$$VMR = \frac{\mu_0 N_0}{kL} \times \frac{E}{N^2} \times \frac{I_{RH^+} \times TR_{H_3O^+}}{I_{H_3O^+} \times TR_{RH^+}} \qquad (1)$$

where,

VMR - the volume mixing ratio, ppbv;

$\mu_0$ - the reduced mobility, cm V$^{-1}$ s$^{-1}$;

$N_0$ - the gas number density at standard pressure (1 atm) and temperature (273.15 K);

k - the reaction rate coefficient, $10^{-9}$ cm$^3$ s$^{-1}$;

L - the length of the drift tube, 9.3 cm in this study;

E - the electric field strength, V m$^{-1}$;

N - the air density in the drift tube, m$^{-3}$;

I - the numbers of detected ions, cps;

TR - the transmission factor of respective ions."

Page 5, lines 28-30, discussion relating to sensitivity dependence of HCHO to relative humidity: the authors should acknowledge and discuss similar work performed on the same VOC standard using the same equipment, in which similar observations were made, namely: Beauchamp et al. Meas. Sci. Technol., 24, 125003, 2013. How do the present measurements compare to those reported in the aforementioned article?

Response: thanks for the excellent comments. The reference was added in the revised manuscript. And this part was expanded as follows.

"Beauchamp et al. (2013) used a similar mixed gas standard for PTR-MS calibration, low sensitivity of HCHO was also found below 2 ncps ppbv-1 when the RH ranged from 20% to 100%. Moreover, strong nonlinear dependence on RH for HCHO was found by Beauchamp et al. (2013), and the sensitivity of HCHO significantly decreased by about 50% when the RH of inlet air increased from 20% to 100%, which is comparable to our study."

Page 9, lines 3-4: the authors start this section by referring to a comparison between PTR-MS and DNPH-HPLC data for acetone and propionaldehyde, but in the next sentence discuss other compounds and an unrelated figure. The discussion on the former reappear at the end of this paragraph but with no presentation of the data: are the data for acetone not shown? Consider repositioning the discussion on those data to the start of the paragraph and indicate that the data are not presented.

Response: thanks for the good suggestions. This part was revised in our manuscript as follows.

"As shown in Fig. 9, good linear correlations were found for formaldehyde, acetaldehyde and MEK with correlation coefficients of 0.79, 0.75, 0.93, respectively, and with slopes of $1.00 \pm 0.10$, $1.10 \pm 0.33$, and $0.88 \pm 0.058$, respectively. For acetone, the slope is $0.76 \pm 0.23$ (PTR-MS to HPLC) with a correlation coefficient of 0.60."

"Wisthaler et al. (2008) reported the inter-comparison between PTR-MS and DNPH-HPLC in an atmosphere simulation chamber, good agreement was found between PTR-MS and DNPH-HPLC while ambient air was introduced into the chamber, but the

concentration of HCHO measured by DNPH-HPLC was less than by PTR-MS, which could be caused by some interferences for DNPH-HPLC method or the varying performance of the KI ozone scrubber. Overestimation of DNPH-HPLC for HCHO in the presence of NO2 was also reported by Herrington and Hays (2012), because NO can be oxidized to NO2 in the upstream ozone scrubber, and NO2 will react with DNPH to form 2,4-dinitrophenylazide (DNPA), which has the similar chromatographic properties with the formaldehyde-DNP-hydrazone. Hence, the intercept of -0.03 for HCHO inter-comparison between PTR-MS and DNPH-HPLC in this study may be explained by the interference of NO2 because of the high NOx levels at the roadside sampling site."

"Low acetaldehyde collection efficiencies (CEs), ranging from 1 to 62% was found by Herrington et al. (2007) for the typical 24-hour sampling period which can lead to the underestimation of acetaldehyde by DNPH-HPLC method. And this artifact is consistent with the result for acetaldehyde inter-comparisons in this study. It was found that ketone concentrations determined by DNPH-HPLC method could be underestimated by 35% ~ 80% under high RH (>50%) condition when the temperature is about 22$^\circ$C (Ho et al., 2014). This DNPH issue could explain the 12% difference between PTR-MS and DNPH-HPLC for MEK and the relative bad agreement for acetone in our study."

Figures 9 and 11: where are the error bars for the DNPH-HPLC and off-line GC-MSD/FID/ECD analyses, or is there no measurement error associated with these systems?

Response: thanks for the excellent comments. The error bar of the y-axis parameter stands for the standard deviation of 24-hour averaged PTR-MS data as stated in Figure 9 and Figure 12 (in the revised manuscript). But both DNPH cartridge samples and canister samples were collected once during 24 hours. So the standard deviation was not existed for DNPH cartridge and canister sample. Therefore, only error bar of the y-axis parameter was plotted.

Figure 11 and 12 captions: the authors should elaborate in the caption on what the "corrected" data are. Errors.

Response: thanks for the suggestions. It has been revised in our manuscript as follows.

"Figure 11. Inter-comparison between ambient aromatic hydrocarbon measurements by PTR-MS and by on-line GC-FID during the field study at MK in Hong Kong: (A) benzene, (B) toluene, and (C) C2-benzenes. Original data are plotted in black dots, grey dots represent **the corrected data based on Eq. (6) and Eq. (7)**. Linear regression fits for original and corrected data are indicated by the solid black line and the solid grey line respectively. Dashed line is the 1:1 line for reference."

"Figure 12. Inter-comparison between ambient aromatic hydrocarbon measurements by PTR-MS (**corrected based on Eq. (6) and Eq. (7)**) and by off-line canister measurements using GC-MSD/FID/ECD during the field study at MK in Hong Kong: (A) benzene, (B) toluene, and (C) C2-benzenes. Linear regression fits are indicated by the solid black line. Error bar stands for the standard deviation of 24-hour averaged PTR-MS data. Dashed line is the 1:1 line for reference."

Throughout: Please be consistent in the use of VOC and VOCs for singular and plural, respectively (similarly for OVOC).

Response: thanks for the comments. It has been revised in the manuscript.

Abstract, line 24 and p2, line 12: place parentheses around ECD and not electron capture detection.

Response: thanks for the comments. It has been revised in the manuscript.

Page 2, line 29: change to "HCHO can be protonated by the following reaction".

Response: thanks for the comments. It has been revised in the manuscript.

Page 2, line 32: change "is just slight higher" to "is just slightly higher".

Response: thanks for the comments. It has been revised in the manuscript.

Page 7, line 22: "C2-benzenes" not "C2-benzens". Page 9, line 24: "field" not "filed" sampling study.

Response: thanks for the comments. It has been revised in the manuscript.

Figure 1 caption: should be "dashed lines" not "das lines".

Response: thanks for the comments. It has been revised in the manuscript.

References

de Gouw, J., and Warneke, C.: Measurements of volatile organic compounds in the earth's atmosphere using proton-transfer-reaction mass spectrometry, Mass spectrometry reviews, 26, 223-257, 10.1002/mas.20119, 2007.

Salisbury, G., Williams, J., Holzinger, R., Gros, V., Mihalopoulos, N., Vrekoussis, M., Sarda-Estève, R., Berresheim, H., von Kuhlmann, R., Lawrence, M., and Lelieveld, J.: Ground-based PTR-MS measurements of reactive organic compounds during the MINOS campaign in Crete, July–August 2001, Atmos. Chem. Phys., 3, 925-940, 10.5194/acp-3-925-2003, 2003.

---

## Author Comment (AC2) · 24 Aug 2016

Cui et al. show VOC measurements from PTR-MS in Hong Kong. They also discuss humidity dependence of formaldehyde in PTR-MS and inter-comparison with several other techniques, including DNPH, canister samples, online GC-FID. However, this manuscript generally does not provide much new information and new technique. Many valuable experiences from over 20 years of work in the PTR-MS community are not fully reflected in the data processing procedures in this manuscript. The inter-comparison results are not as good as those previously reported in the literature, but the authors do not provide good reasons about it. Thus, I do not think this manuscript is suitable to publish in AMT, unless this manuscript is totally re-written and provide more information that can support new idea/technique.

Response: we thank the reviewer very much for the critical comments which would help us to improve our work. The major concerns of the reviewer are on (1) the significance of this study and (2) the weakness of the discussion part about inter-comparisons between PTR-MS and other techniques or even other previous studies. Below we first address these major concerns and then reply individually the specific comments. For clarity, the reviewer's comments are listed below in black italics, while our responses and changes in manuscript are shown in blue and red, respectively.

**(1) On the significance of this study**

We agree with the reviewer that the present study is based on the PTR-MS technique, which has been studied worldwide through the past 20 years. Here we just would like to state the significance of our study.

Indeed, PTR-MS is not a brand new technique for measuring OVOC and VOC, and the other techniques used in this study also have a relative long history. But very few studies used PTR-MS at roadside microenvironment for OVOC and VOC measurements, **even less studies has been conducted at a sampling site with**

**extremely huge traffic volume as in our study**, because the traffic density of Hong Kong is one of the largest worldwide, and the Mong Kok site which was selected as the sampling location has the largest traffic volume in Hong Kong. Besides, **it was the first time that PTR-MS was used in Hong Kong to measure ambient OVOC and VOC.** So it is very necessary and important to do the inter-comparison between PTR-MS and other techniques in an urban roadside microenvironment of Hong Kong.

Moreover, few studies have conducted inter-comparisons between PTR-MS and alternative methods for OVOCs. **Even fewer studies have investigated inter-comparison between PTR-MS and DNPH-HPLC in urban roadside areas** as stated in "Section 3.5" of our manuscript. Because only DNPH-HPLC method has been used for measuring ambient OVOCs in Hong Kong, and there is no on-line techniques (e.g. DOAS instrument and Hantzsch monitor) have been used in Hong Kong before and recently. It is more necessary to conduct the inter-comparison between PTR-MS and DNPH-HPLC for OVOCs in Hong Kong. Our study firstly investigated the influence of relative humidity and temperature of inlet air for OVOC and VOC measurements separately. And better agreement was found between PTR-MS and DNPH-HPLC for formaldehyde than previous other studies by using the simulation model of this study.

**Overall, although there is no brand new technique for OVOC and VOC measurement in this study, this study provide some new insights into the relative humidity and temperature dependence for PTR-MS measurement, and some useful information of the inter-comparison between PTR-MS and other techniques in an urban roadside microenvironment in Hong Kong.** These results should be useful for OVOC and VOC measurement by PTR-MS in other urban roadside areas with high traffic volume of the world.

**(2) On the weakness of the discussion part**

Actually, not all of the inter-comparison results are not as good as those previously reported in the literature, especially for HCHO measurement by using the simulation model firstly raised by this study. Moreover, if only inter-comparisons for OVOC and VOC in urban areas are conducted, we can easily find that the inter-comparison results

are almost all close to other studies when the same techniques were used. We agree with the reviewer that more discussions should be provided to illustrate the difference or similarity with other studies. Although we have discussed the comparisons with other studies in "Section 3.5" of the original manuscript, we accept the comments and suggestions of the reviewer to expand more in our revised manuscript in "Section 3".

Specific comments

1. Hong Kong is just one city in Pearl River Delta region. Thus, the authors might want to introduce more previous work in PRD region, rather than just Hong Kong. Several researchers have reported PTR-MS results in PRD region [Wang et al., 2016], and more in other parts of China, but the authors do not acknowledge these references in the introduction.

Response: thanks for the good comments and suggestions. The reference was added in the revised manuscript. And related information was added in the revised manuscript as follows.

"PTR-MS has been used in China for environmental studies throughout these years (Wang et al., 2014; Wang et al., 2016). But PTR-MS was firstly used in an urban roadside microenvironment of Hong Kong to measure ambient OVOCs and VOCs in this study."

2. Formaldehyde humidity dependence in PTR-MS. Several papers have discussed this issue previously [Inomata et al., 2008; Vlasenko et al., 2010; Warneke et al., 2011]. Especially, Valsenko et al. derived explicit equations to fit the sensitivity vs. absolute humidity, based on the equilibrium between forward and backward reactions. In this manuscript, the authors tried to fit the observed rate coefficient (k) and sensitivity with relative humidity, temperature and absolute humidity. All of these tests are just using one dataset with different parameters/equations to derive curves for the correction, but without knowing the physical meaning of parameters/equations. The best way to do it should use the equations shown in Valsenko et al. Thus, Figure 2-8 should be replaced with one Figure similar to Fig. 5 in Valsenko et al.

Response: thanks for the comments. Several previous studies have discussed the humidity dependence of formaldehyde either discussed the sensitivity variance against relative humidity only or against absolute humidity. But temperature also lead to the sensitivity variance in our study. No matter RH&T or absolute humidity is chosen to do the simulation, the sensitivity variance of formaldehyde depends on the amount of water vapor concentration in ambient air, because the proton affinity (PA) of formaldehyde is just slightly higher than that of water, so the backward reaction of protonated formaldehyde with $H_2O$ is not negligible. Moreover, the simulation model raised in this study could offer a better agreement for formaldehyde than using the fitting results by absolute humidity. It is easier to measure RH and T of ambient air for further correction of formaldehyde results by PTR-MS in practice.

3. PTR-MS operations: how often the instrument is calibrated, how is background determined, how often do you do background, how often the humidity dependence curve for formaldehyde is conducted (just one in 2 years?).

Response: the instrument was calibrated every six days, and the background level was determined by using the zero air generated by a Gas Calibration Unit (GCU) (IONICON Analytik GmbH, Innsbruck, Austria) for half-hour each day, and the humidity dependence curve for formaldehyde was conducted every month. Details can be found in the revised manuscript as follows.

"Zero air was generated by a Gas Calibration Unit (GCU) (IONICON Analytik GmbH, Innsbruck, Austria) with a VOC-scrubber installed inside the GCU. Background level was determined by using the zero air for half-hour each day. The relative humidity and temperature of inlet gas were set by adjusting the humidification chamber and a dew point mirror inside the GCU. Ionimed mix-VOCs gas standard was diluted with four different ratios (0.08, 0.06, 0.04 and 0.001) to calibrate the PTR-MS. The accuracy and the measurement precision of the PTR-MS was 20% and 10%, respectively. The accuracy of PTR-MS measurement was dependent on the accuracy of the reaction rate coefficient, fragmentation, gas standard and flow rate (Salisbury et al., 2003;Kim et al., 2009). The precision was determined based on the standard deviation of the background

signal at each mass during 5-min average measurement for each specie. Calibrations were done every six days for ensuring the accuracy of PTR-MS."

4. PTR-MS data processing: how signal is normalized, do you see sensitivity drift with time, how background is interpolated (especially formaldehyde). The authors should apply the widely accepted data reduction methods shown in many previous papers (most important ones [de Gouw and Warneke, 2007; de Gouw et al., 2003]).

Response: the sensitivity was defined as the signal of $RH^+$ ions obtained at a VMR of 1 ppbv and normalized to a standard $H_3O^+$ signal ($I_{H3O+}$) of $10^6$ counts $s^{-1}$ as presented in many previous studies (de Gouw et al., 2003;de Gouw and Warneke, 2007). The sensitivity only drifted a little during the whole sampling period. The background was determined by introducing the zero air generated by GCU, and the background level varied within the difference less than 5% for all species, the monthly averaged background level was used during each sampling month.

5. Inter-comparison: In addition to PTR-MS, the QA/QC procedures should be provided to evaluation the data quality. Previous studies also found many limitations/interferences for DNPH method, such as formaldehyde [Wisthaler et al., 2008], aldehydes [Herrington and Hays, 2012] and ketones [Ho et al., 2014]. Without acknowledging these interferences, the inter-comparison is meaningless. From Figure 9 in the manuscript, the bad agreement for acetone might be a DNPH issue.

Response: thanks for the kind comments and suggestions. The mentioned references were added and the related part was expand in the revised manuscript as follow.

"Wisthaler et al. (2008) reported the inter-comparison between PTR-MS and DNPH-HPLC in an atmosphere simulation chamber, good agreement was found between PTR-MS and DNPH-HPLC while ambient air was introduced into the chamber, but the concentration of HCHO measured by DNPH-HPLC was less than by PTR-MS, which could be caused by some interferences for DNPH-HPLC method or the varying performance of the KI ozone scrubber. Overestimation of DNPH-HPLC for HCHO in the presence of NO2 was also reported by Herrington and Hays (2012), because NO

can be oxidized to NO2 in the upstream ozone scrubber, and NO2 will react with DNPH to form 2,4-dinitrophenylazide (DNPA), which has the similar chromatographic properties with the formaldehyde-DNP-hydrazone. Hence, the intercept of -0.03 for HCHO inter-comparison between PTR-MS and DNPH-HPLC in this study may be explained by the interference of NO2 because of the high NOx levels at the roadside sampling site."

"Low acetaldehyde collection efficiencies (CEs), ranging from 1 to 62% was found by Herrington et al. (2007) for the typical 24-hour sampling period which can lead to the underestimation of acetaldehyde by DNPH-HPLC method. And this artifact is consistent with the result for acetaldehyde inter-comparisons in this study. It was found that ketone concentrations determined by DNPH-HPLC method could be underestimated by 35 ~ 80% under high RH (>50%) condition when the temperature is about 22 $^{\circ}$C (Ho et al., 2014). This DNPH issue could explain the 12% difference between PTR-MS and DNPH-HPLC for MEK and the relative bad agreement for acetone in our study."

6. Eq. 7: If you use m107 to calibrate C2 benzenes, then C2 benzenes should not be corrected. Where is 0.2235 from? Note that only ethylbenzene fragments significantly. Thus, Eq. 6 should be [Benzene]=[m79]-0.2235*[m107]*f. f is the fraction of ethylbenzene in ethylbenzene+xylenes, which can be from GC-FID measurements. Then, PTR-MS measurements would somewhat rely on GC-FID measurements. How would this reliance affect inter-comparison?

Response: thanks for the comments. Since higher aromatics (ethylbenzene and propylbenzene) can start to fragment into ions at 79 amu (de Gouw and Warneke, 2007), this fragmentation could lead to the overestimation of benzene at 79 amu and underestimation of C2-benzenes at 107 amu (Rogers et al., 2006).

($S_{Ethylbenzene}$ / $S_{Benzene}$) $BF_{Ethylbenzene}$ [Ethylbenzene] is the fragment from ethylbenzene to 79 amu. In this study, ($S_{Ethylbenzene}$ / $S_{Benzene}$) = 1.58, $BF_{Ethylbenzene}$ = 32%, [Ethylbenzene] = 0.3615 [C$_2$-Benzenes], where 0.3615 was the fraction of ethylbenzene in C$_2$-Benzenes.

Thus, 0.1827 {C₂-Benzenes] was the fragment from ethylbenzene to 79 amu. Then, [C₂-Benzenes] = [M107] + 0.1827 {C₂-Benzenes], so [C₂-Benzenes] = 1.2235 {M107]. [Benzene] = [M79] – 0.1827 {C₂-Benzenes] = [M79] – 0.1827 ·1.2235 {M107] = [M79] – 0.2235 [M107]. In this case, PTR-MS would somewhat rely on GC-FID measurements or canister results, **because the PTR-QMS used in this study cannot differentiate isomers. And this is really the limitation of PTR-QMS. But this is a better way to get a more reliable results if the fraction of ethylbenzene in C₂-Benzenes can be obtained from other techniques (e.g. GC-FID).**

7. Figure 1 should only contain the periods with only PTR-MS measurements. The time series of VOCs should be also shown.

Response: thanks for the comments. Figure 1 was used to explain why we set the air condition at RH = 80% and T = 25°C for calibration. In order to present the time series of VOCs measured by PTR-MS, Figure. 10 was added in the revised manuscript as follows.

"Time series of benzene, toluene and C2-benzenes results obtained by on-line GC-FID and PTR-MS during the sampling period were plotted in Fig. 10."

[Figure]

"Figure 10. Measurement results for benzene, toluene and C2-benzenes obtained during

the field study at MK in Hong Kong. The black lines show the on-line GC-FID data, and the green lines show the PTR-MS results."

References

5    de Gouw, J., and Warneke, C.: Measurements of volatile organic compounds in the earth's atmosphere using proton-transfer-reaction mass spectrometry, Mass spectrometry reviews, 26, 223-257, 10.1002/mas.20119, 2007.

de Gouw, J. A., Goldan, P. D., Warneke, C., Kuster, W. C., Roberts, J. M., Marchewka, M., Bertman, S. B., Pszenny, A. A. P., and Keene, W. C.: Validation of proton transfer reaction-mass spectrometry (PTR-MS) measurements of gas-phase organic compounds in the atmosphere during the New England Air Quality Study (NEAQS) in 2002, Journal of Geophysical Research: Atmospheres, 108, 10.1029/2003jd003863, 2003.

Rogers, T. M., Grimsrud, E. P., Herndon, S. C., Jayne, J. T., Kolb, C. E., Allwine, E., Westberg, H., Lamb, B. K., Zavala, M., Molina, L. T., Molina, M. J., and Knighton, W. B.: On-road measurements of volatile organic compounds in the Mexico City metropolitan area using proton transfer reaction mass spectrometry, International Journal of Mass Spectrometry, 252, 26-37, 10.1016/j.ijms.2006.01.027, 2006.

---

## Author Comment (AC3) · 24 Aug 2016

The authors have applied a suite of techniques to measure VOC and OVOC at a roadside site in Hong Kong and have compared the data to those generated by a PTR-MS. Measurements of formaldehyde are examined in detail and a method has been developed to address the RH and T dependence of the calibration for future studies. This approach seems reasonable. OVOCs measured by PTR-MS are compared with a DNPH method and aromatics measured by PTR-MS are compared to in-situ GC-FID and canister samples. Besides my question regarding the fitting method below, the comparison has been done well enough, very much along the line of previous groups.

Response: we appreciate the reviewer for the positive comments and helpful suggestions. The manuscript has been revised and improved based on these suggestions. For clarity, the reviewer's comments are listed below in black italics, while our responses and changes in manuscript are shown in blue and red, respectively.

However, I cannot help feeling though that there is an opportunity missed here. Surely the GC-FID also measured the OVOC (i.e. not just the aromatics)? Can their peaks not be calibrated by carbon number and compared with the PTR-MS/DNPH/can methods as well? The result would be an assessment/validation of the in-situ GC-FID measurement of OVOC which could be of use to the group in future measurement campaigns where the PTR-MS is not available. Moreover, do the canister data not deliver values of acetone, propanal and acetaldehyde for comparison? This again would be an interesting comparison extension to this study. There has been much discussion of potential canister artifacts previously but they may compare well under the conditions where ozone is low.

Response: many thanks for the comments. We understand that it will be better if other techniques (on-line GC-FID and canister samples) can measure OVOC. Unfortunately,

both on-line GC-FID and canister samples cannot measure OVOC in this study. Only less than 30 $C_2$-$C_9$ NMHC species without any OVOC species were measured by the on-line GC-FID as previous studies in Hong Kong (Xue et al., 2014;Ou et al., 2015). The canister data covered a wide range of NMHC species (C2−C10; >50 species), but these species only contained alkanes, alkynes, aromatics, halocarbons and sulphur compounds as stated by Colman et al. (2001). Hence, **OVOC can only be measured by PTR-MS and DNPH-HPLC methods in this study.** But it is a good suggestion that we will try to measure more OVOC species by different techniques in our future field campaigns.

One of the greatest problems in sampling and measuring OVOC will be ozone as it can make OVOC (and remove reactive alkenes) when trapped in canisters or concentrators. If ozone was measured at this site is would be very interesting to compare the degree of fit with ozone levels. Most city roadside sites show low ozone, due to the titration of NO, but in the afternoon it is likely that ozone levels will increase as photochemical production and down-mixing kick in. This would mean correlation could be expected to deteriorate later in the afternoon if ozone reactions become important. Examining this influence of local ozone of the quality of fit would be a very interesting addition to this paper. It is likely to impact the GC and DNPH methods more than the PTR-MS.

Response: thanks for the suggestions. A temperature controlled copper tube coated with KI is used as the ozone scrubber in the ATEC Model 2200 automated sampler (Model 2200, Malibu, CA) to remove ozone during carbonyl sampling with DNPH cartridges in this study. Additionally, an ozone scrubber (Sep-Pak; Waters Corporation, Milford, MA) was also connected to the DNPH cartridge for each sample. It has been reported widely that the artifacts can be eliminated by installing the ozone scrubber upstream, and the oxidants at ambient level can be removed efficiently (Rodier et al., 1993;Sirju and Shepson, 1995;Helmig, 1997).

Moreover, the ozone level at the sampling site during the sampling period was quite low (less than 10 ppbv), and the maximum ozone level was less than 40 ppbv even in the afternoon because of the high traffic volume at the sampling site. According to Sirju

and Shepson (1995)'s study, ozone concentration below 40 ppbv is a typical value of clean ambient air, and using KI trap is effective to remove ozone for both urban and rural areas. Besides, ozone removal efficiency >99% was measured for the scrubber with ozone levels of 100 ppb in earlier studies. Our study also showed good correlations for formaldehyde and acetaldehyde between the PTR-MS and DNPH-HPLC method. Hence, ozone is not the major interference for OVOC measurement by DNPH-HPLC method. The relative bad agreement for acetone might be a DNPH issue but not ozone interference (Ho et al., 2014). In the revised manuscript, more information of ozone scrubber usage and the inter-comparison part was expanded as follows.

"An ozone scrubber (Sep-Pak; Waters Corporation, Milford, MA) was used to remove ozone during carbonyl sampling with 2,4-dinitrophenylhydrazine (DNPH) cartridges (Waters Sep-Pak DNPH-silica, Milford, MA)."

"Wisthaler et al. (2008) reported the inter-comparison between PTR-MS and DNPH-HPLC in an atmosphere simulation chamber, good agreement was found between PTR-MS and DNPH-HPLC while ambient air was introduced into the chamber, but the concentration of HCHO measured by DNPH-HPLC was less than by PTR-MS, which could be caused by some interferences for DNPH-HPLC method or the varying performance of the KI ozone scrubber. Overestimation of DNPH-HPLC for HCHO in the presence of $NO_2$ was also reported by Herrington and Hays (2012), because NO can be oxidized to $NO_2$ in the upstream ozone scrubber, and $NO_2$ will react with DNPH to form 2,4-dinitrophenylazide (DNPA), which has the similar chromatographic properties with the formaldehyde-DNP-hydrazone. Hence, the intercept of -0.03 for HCHO inter-comparison between PTR-MS and DNPH-HPLC in this study may be explained by the interference of $NO_2$ because of the high $NO_x$ levels at the roadside sampling site."

For the correlation plots a simple y=mx+c form is used. This assumes that the x-axis parameter is correct and without error. More appropriate in this case would be to use orthogonal distance regression to account for error in both axes, since the DNPH method will also contain errors to some degree.

Response: thanks for the excellent comments. Firstly, the linear regression is used by many previous studies for comparison analysis (Warneke et al., 2001;de Gouw and Warneke, 2007;Warneke et al., 2011;Jobson et al., 2010;Wang et al., 2014;Kuster et al., 2004;Kato et al., 2004). Because "Section 3.5 - comparison with other studies" is one of the key parts in this manuscript, it is better to choose the same analytical method to conduct the comparison for those parameters (namely slope, intercept and correlation coefficient). Secondly, the error bar of the y-axis parameter stands for the standard deviation of 24-hour averaged PTR-MS data as stated in Figure 9 and Figure 12. But both DNPH cartridge samples and canister samples were collected once during 24 hours. So the standard deviation was not existed for DNPH cartridge and canister sample. Therefore, only error bar of the y-axis parameter was plotted.

The PTR-MS accuracy is given as 20% and the precision as "about 10%". Since this paper is an instrumental comparison I think it is necessary to expand on this and explain where the 20% comes from and the measurement precision of each species. Likewise the GC-FID accuracy and precision is given as the same as for the PTR-MS, but without explanation. How these figures arrived at should be given in more detail.

Response: thanks for the useful comments and suggestions. We agree the point of the reviewer, and we expand on the methodology part and give more explanation in the revised manuscript. Several parameters (reaction rate coefficient, fragmentation, flow rate, gas standard…) lead to accuracy, most important is the reaction rate coefficient (Salisbury et al., 2003). One important part of our study was to study the experimental reaction rate coefficient, and it also point out the significance of our study. The related information were added in the manuscript as follows.

"The accuracy and the measurement precision of the PTR-MS was 20% and 10%, respectively. The accuracy of PTR-MS measurement was dependent on the accuracy of the reaction rate coefficient, fragmentation, gas standard and flow rate (Salisbury et al., 2003;Kim et al., 2009). The precision was determined based on the standard deviation of the background signal at each mass during 5-min average measurement for each

specie."

"The accuracy was based on weekly span checks and monthly calibration. The precision was based on the 95% probability limits for the integrated precision check results (Ling et al., 2013;Lyu et al., 2016)."

The potential interference of ozone may be an explanation of the relatively poor fit of PTR-MS vs DNPH for acetaldehyde (and also acetone) in figure 9 (D and E). It might be illuminating to color the points as a function of daily average ozone (if available).

Response: thanks for the kind comments. Actually, ozone is not the key issue for the
10 relative poor fit of PTR-MS vs DNPH for acetaldehyde (and also acetone) as stated in the third response. The relatively poor fit is mainly caused by the DNPH issue for ketones and low collection efficiencies for acetaldehyde because of the long time sampling period for DNPH cartridge samples. More detailed explanation for the inter-comparison between PTR-MS and DNPH-HPLC was expanded in the revised
15 manuscript as follows.

"Low acetaldehyde collection efficiencies (CEs), ranging from 1 to 62% was found by Herrington et al. (2007) for the typical 24-hour sampling period which can lead to the underestimation of acetaldehyde by DNPH-HPLC method. And this artifact is consistent with the result for acetaldehyde inter-comparisons in this study. It was found
20 that ketone concentrations determined by DNPH-HPLC method could be underestimated by 35 ~ 80% under high RH (>50%) condition when the temperature is about 22 $^\circ$C (Ho et al., 2014). This DNPH issue could explain the 12% difference between PTR-MS and DNPH-HPLC for MEK and the relative bad agreement for acetone in our study."

25

Minor points. Introduction, line 4. Perhaps a more relevant reference concerning the human health impacts would be Lelieveld et al. (doi:10.1038/nature15371).

Response: thanks for the comments. The reference was added in the revised manuscript.

30

"Special attention has been paid on the characteristics" should be "paid to".

Response: thanks for the comments. It has been revised in the manuscript.

Section 2.2, give details of the particulate filter (i.e. material, pore size, how often changed).

Response: thanks for the comments. Details of the particulate filter is added in the revised manuscript as follows.

"An in-line particulate filter (4.7 mm Teflon-membrane filter assembly, Whatman Inc., Clifton, NJ,USA) was used to prevent particles from entering the instrument."

Section 2.4 line 20. What were these strict QA/QC procedures? If necessary give the reference where they are detailed directly afterwards.

Response: thanks for the comments. References were added afterwards and the manuscript was revised as follows.

"The accuracy was based on weekly span checks and monthly calibration. The precision was based on the 95% probability limits for the integrated precision check results (Ling et al., 2013; Lyu et al., 2016)."

Section 3.3. line 2, was glyoxal measured, perhaps it can contribute to mass 59?

Response: thanks for the comments. Glyoxal was not measured in this study, and acetone occupies the most (90 ~ 100%) at mass 59 (de Gouw and Warneke, 2007). So glyoxal did not affect the acetone measurement significantly by PTR-MS. But it is good suggestions that we will explore the influence of glyoxal on acetone measurement by PTR-MS in the future.

Section 3.3 line 3 "benzens" should be benzenes.

Response: thanks for the comments. It has been revised in the manuscript.

Figure 8 Xaxis label is misspelt "Measuremeasured".

Response: thanks for the comments. It has been revised in the manuscript.

Conclusions, line 24, filed should be field.

Response: thanks for the comments. It has been revised in the manuscript.

References

Colman, J. J., Swanson, A. L., Meinardi, S., Sive, B. C., Blake, D. R., and Rowland, F. S.: Description of the Analysis of a Wide Range of Volatile Organic Compounds in Whole Air Samples Collected during PEM-Tropics A and B, Analytical Chemistry, 73, 3723-3731, 10.1021/ac010027g, 2001.

de Gouw, J., and Warneke, C.: Measurements of volatile organic compounds in the earth's atmosphere using proton-transfer-reaction mass spectrometry, Mass spectrometry reviews, 26, 223-257, 10.1002/mas.20119, 2007.

Helmig, D.: Ozone removal techniques in the sampling of atmospheric volatile organic trace gases, Atmospheric Environment, 31, 3635-3651, http://dx.doi.org/10.1016/S1352-2310(97)00144-1, 1997.

Ho, S. S. H., Chow, J. C., Watson, J. G., Ip, H. S. S., Ho, K. F., Dai, W. T., and Cao, J.: Biases in ketone measurements using DNPH-coated solid sorbent cartridges, Analytical Methods, 6, 967, 10.1039/c3ay41636d, 2014.

Jobson, B. T., Volkamer, R. A., Velasco, E., Allwine, G., Westberg, H., Lamb, B. K., Alexander, M. L., Berkowitz, C. M., and Molina, L. T.: Comparison of aromatic hydrocarbon measurements made by PTR-MS, DOAS and GC-FID during the MCMA 2003 Field Experiment, Atmos. Chem. Phys., 10, 1989-2005, 10.5194/acp-10-1989-2010, 2010.

Kato, S., Miyakawa, Y., Kaneko, T., and Kajii, Y.: Urban air measurements using PTR-MS in Tokyo area and comparison with GC-FID measurements, International Journal of Mass Spectrometry, 235, 103-110, http://dx.doi.org/10.1016/j.ijms.2004.03.013, 2004.

Kuster, W. C., Jobson, B. T., Karl, T., Riemer, D., Apel, E., Goldan, P. D., and Fehsenfeld, F. C.: Intercomparison of Volatile Organic Carbon Measurement Techniques and Data at La Porte during the TexAQS2000 Air Quality Study,

Environmental Science & Technology, 38, 221-228, 10.1021/es034710r, 2004.

Ou, J., Guo, H., Zheng, J., Cheung, K., Louie, P. K. K., Ling, Z., and Wang, D.: Concentrations and sources of non-methane hydrocarbons (NMHCs) from 2005 to 2013 in Hong Kong: A multi-year real-time data analysis, Atmospheric Environment, 103, 196-206, http://dx.doi.org/10.1016/j.atmosenv.2014.12.048, 2015.

Rodier, D. R., Nondek, L., and Birks, J. W.: Evaluation of ozone and water vapor interferences in the derivatization of atmospheric aldehydes with dansylhydrazine, Environmental Science & Technology, 27, 2814-2820, 10.1021/es00049a022, 1993.

Salisbury, G., Williams, J., Holzinger, R., Gros, V., Mihalopoulos, N., Vrekoussis, M., Sarda-Estève, R., Berresheim, H., von Kuhlmann, R., Lawrence, M., and Lelieveld, J.: Ground-based PTR-MS measurements of reactive organic compounds during the MINOS campaign in Crete, July–August 2001, Atmos. Chem. Phys., 3, 925-940, 10.5194/acp-3-925-2003, 2003.

Sirju, A.-P., and Shepson, P. B.: Laboratory and Field Investigation of the DNPH Cartridge Technique for the Measurement of Atmospheric Carbonyl Compounds, Environmental Science & Technology, 29, 384-392, 10.1021/es00002a014, 1995.

Wang, M., Zeng, L., Lu, S., Shao, M., Liu, X., Yu, X., Chen, W., Yuan, B., Zhang, Q., Hu, M., and Zhang, Z.: Development and validation of a cryogen-free automatic gas chromatograph system (GC-MS/FID) for online measurements of volatile organic compounds, Anal. Methods, 6, 9424-9434, 10.1039/c4ay01855a, 2014.

Warneke, C., van der Veen, C., Luxembourg, S., de Gouw, J. A., and Kok, A.: Measurements of benzene and toluene in ambient air using proton-transfer-reaction mass spectrometry: calibration, humidity dependence, and field intercomparison, International Journal of Mass Spectrometry, 207, 167-182, http://dx.doi.org/10.1016/S1387-3806(01)00366-9, 2001.

Warneke, C., Veres, P., Holloway, J. S., Stutz, J., Tsai, C., Alvarez, S., Rappenglueck, B., Fehsenfeld, F. C., Graus, M., Gilman, J. B., and de Gouw, J. A.: Airborne formaldehyde measurements using PTR-MS: calibration, humidity dependence, inter-comparison and initial results, Atmospheric Measurement Techniques, 4, 2345-2358, 10.5194/amt-4-2345-2011, 2011.

Xue, L., Wang, T., Louie, P. K. K., Luk, C. W. Y., Blake, D. R., and Xu, Z.: Increasing External Effects Negate Local Efforts to Control Ozone Air Pollution: A Case Study of Hong Kong and Implications for Other Chinese Cities, Environmental Science & Technology, 48, 10769-10775, 10.1021/es503278g, 2014.